# Transient Presence of Live *Leptospira interrogans* in Murine Testes

Advait Shetty,[a] Suman Kundu,[b] Frédérique Vernel-Pauillac,[c] Gwendoline Ratet,[c] Catherine Werts,[c] Maria Gomes-Solecki[a,b]

[a]Department of Pharmaceutical Sciences, The University of Tennessee Health Science Center, Memphis, Tennessee, USA
[b]Department of Microbiology, Immunology and Biochemistry, The University of Tennessee Health Science Center, Memphis, Tennessee, USA
[c]Institut Pasteur, Université de Paris, CNRS UMR6047, INSERM U1306, Unité de Biologie et Génétique de la paroi bactérienne, Paris, France

**ABSTRACT** Analysis of *Leptospira* dissemination and colonization of sex organs in rodents is of significant value as it queries the possibility of mammal-to-mammal venereal transmission. The aim of our study was to evaluate the presence and viability of *Leptospira interrogans* in testes of mice using models of infection that we previously developed. Using sublethal and lethal doses of bioluminescent strains of *L. interrogans* serovars Manilae and Copenhageni, we visualized the presence of leptospires in testes of C57BL/6 mice as early as 30 min and up to days 3–4 postinfection. This was confirmed by qPCR for the Copenhageni serovar after lethal infection of C3H/HeJ mice. In this model, no histopathological changes were noticed in testis. We further studied persistence of serovar Copenhageni in C3H/HeJ testes after lethal and sublethal infection, with different doses of leptospires. No viable leptospires were recovered from testes of lethally infected mice. However, we found live culturable *Leptospira* in testes of 19/19 (100%) sublethally infected mice at the acute phase but not at 15 days postinfection, which corresponds to the chronic phase of renal colonization. The data suggest that colonization of testes with live and potentially infectious leptospires is transient and limited to the spirochetemic phase of infection. Further studies are necessary to evaluate if presence of *Leptospira* in testes of mice leads to excretion in semen and to venereal transmission to female mice.

**IMPORTANCE** Analysis of venereal transmission of *Leptospira* is important to determine if direct animal to animal transmission occurs, which could impact measures to prevent and treat leptospirosis. The goal of this study was to determine if live *Leptospira* colonize mouse testes. We found that colonization of mouse testes with live *Leptospira* was transient and limited to the acute spirochetemic phase of infection and that transient colonization of the testes was insufficient to cause histopathological changes.

**KEYWORDS** *L. interrogans*, *Leptospira*, culture, leptospirosis, lethal infection, testes

Leptospirosis, caused by several pathogenic species of a spirochete named *Leptospira* (1, 2), is a zoonotic infection affecting about 1 million people annually, of which approximately 60,000 perish. The disease is prevalent in regions of Southeast Asia, Africa, and South America and, also, to some extent, in developing countries (2). Slum dwellers are at high risk of exposure to leptospirosis due to substandard sanitation living conditions (2, 3). Rats and mice are asymptomatic reservoir host carriers of this pathogen and shed it in urine. Humans, wild and domestic animals acquire the disease after exposure of abraded skin and mucosa to contaminated water and soil (1, 2, 4).

Previous studies have reported detection of *Leptospira* in reproductive organs of cows and boars (5–7). In females, genital leptospirosis has been linked to stillbirth, abortion and weak progeny (6–9). Detection of leptospiral DNA in vaginal fluid of cows (10) and genital tract of sheep (11, 12) suggested sexual transmission may occur by

Address correspondence to Maria Gomes-Solecki, mgomesso@uthsc.edu.

The authors declare no conflict of interest.

venereal route. In bulls, *L. interrogans* Pomona infection was associated with lesions in testis and epididymis. However, in that particular case the semen may have been contaminated with urine containing *Leptospira* (13). In rams, *Leptospira* was detected in testis and epididymis by culture but no *Leptospira* was microscopically observed (14). In humans, *Leptospira* disseminates through blood to various organs and after colonization of the kidney, *Leptospira* is shed in urine (1, 2, 4). A possible human-to-human sexual transmission case of leptospirosis was reported (15). However, human-to-human transmission of *Leptospira* has not been established.

The goal of this study was to determine if live *Leptospira* colonize mouse testes. Analysis of venereal transmission of *Leptospira* is important to determine if direct animal to animal transmission occurs which could impact measures to prevent and treat leptospirosis.

## RESULTS

**Dissemination of bioluminescent *L. interrogans* in male and female C57BL/6J mice.** We recently showed that male hamsters are more susceptible to lethal infection with *L. interrogans* than females (16). However, it had not been reported in mice, likely because most of the studies were performed in females. Therefore, we infected both male and female albino C57BL/6J mice with $5 \times 10^7$, (a sublethal dose) and $2 \times 10^8$ (a potentially lethal dose) of MFLum1, a bioluminescent derivative of *L. interrogans* serovar Manilae L495. Live imaging was performed 10 min after injection of the luciferin substrate in mice. Only metabolically active live bioluminescent leptospires can glow since emission of bioluminescence requires ATP and oxygen (17). As previously described, at 30 min postinfection (p.i) in female mice, bioluminescent leptospires exiting from the peritoneal cavity were observed concentrated in a round shape (Fig. 1A and B). Then at day 3 p.i, leptospires disseminated in blood, with a complete systemic dissemination in mice infected with the highest dose, and at day 14 p.i leptospires were observed only in kidneys, as expected (17). However, in males we observed a different pattern of dissemination: at 30 min p.i we observed a less defined shape of bioluminescence than in females, and a striking high amount of light in the inguinal area that was not observed in females, and which we hypothesize to be the testes. Although renal colonization was observed at day 14 in 17/18 infected mice, the levels of light in kidneys were rather different between mice within a group. However, no obvious differences were observed between males and females. We also infected black C57BL6/J mice and observed the same differences between females and males harboring MFLum1 in the inguinal area (Fig. 1B).

Then we wondered whether this finding could be extended to other serovars. We repeated the infection in males with $2 \times 10^8$ FGLum4, a bioluminescent derivative of *L. interrogans* Fiocruz L1-130, the parental strain which has been shown to cause more severe leptospirosis in male hamsters. Live imaging was performed every day at the acute phase, and at day 14. Although FGLum4 was not observed at day 14, and therefore is considered as a mutant strain unable to colonize the kidney, we found it in the inguinal area at 30 min p.i, days 1–4 p.i. (Fig. 1C), suggesting that different serovars of *L. interrogans* disseminate and can home to male genitals in the acute phase of leptospirosis.

To further assess the presence of leptospires in testes and study their viability as an indicator of their potential infectivity, we studied and characterized the C3H/HeJ leptospirosis model in males infected with sublethal doses of $10^5$ and $10^6$, and a lethal dose $10^8$ of *L. interrogans* Copenhageni Fiocruz L1-130.

**Weight-loss and survival differences after lethal and sublethal infection of C3H-HeJ mice.** Infection with a sublethal dose of $10^5$ and $10^6$ *L. interrogans* led to survival of all mice (Fig. 2A) while causing significant weight-loss throughout the course of infection until it reached the inflection point on day 11 after which mice started gaining weight. This decline in body weight was proportional with the increase in infection dose (Fig. 2B) and it correlates with the bacteriemia peak measured by qPCR of Leptospiral DNA (days 3–9 postinfection) (Fig. 2C), whereas body weight gains

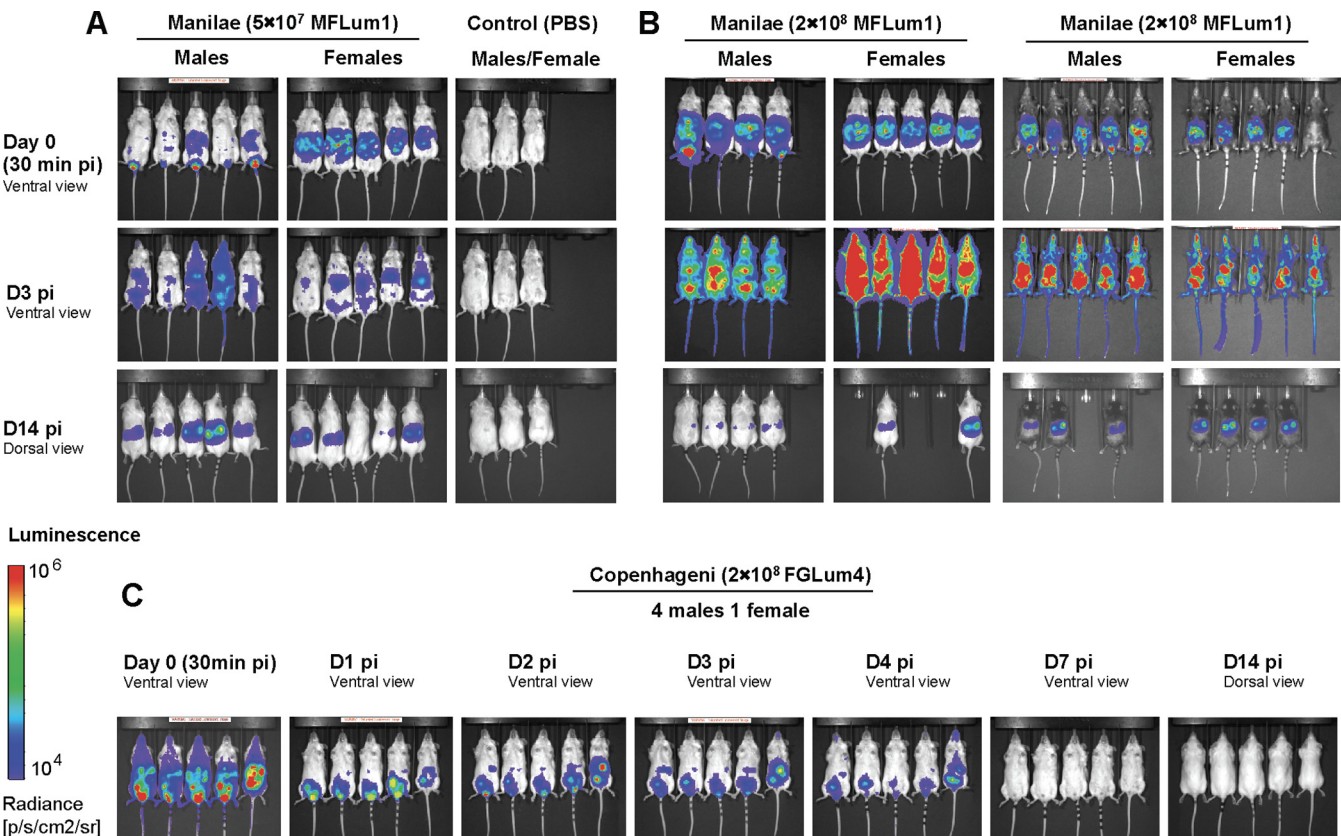

**FIG 1** Live imaging tracking over time of bioluminescent *L. interrogans* injected intra-peritoneally into albino or black, female and male C57BL/6J mice. Live imaging was performed according to (ref FVP 2020) for 5 min, 10 min after the IP administration of d-luciferin, in ventral view (days 0 and 3) and dorsal view (day 14). Data are expressed as radiance of light measured in photons/second/cm$^2$ in and represented by colors, according to the scale, with more intense emitted light in red and less in blue. Pictures of ($n = 5$) male and ($n = 5$) female albino C57BL/6J after infection with $5 \times 10^7$ bioluminescent *L. interrogans* Manilae MFlum1 at Day (D) 0 30 min postinfection (p.i.), day 3 p.i and day 14 p.i, compared to $n = 3$ males and females control mice injected with PBS (A). Pictures of $n = 4$ male and $n = 5$ female Albino C57BL/6J, and $n = 5$ and groups of $n = 5$ black C57BL/6J mice infected with $2 \times 10^8$ MFLUM1 at d0 (30 min), days 3 and 14 p.i. (B). Pictures of $n = 4$ male Albino and $n = 1$ female Albino infected with $2 \times 10^8$ *L. interrogans* Copenhageni FGLum4 at days 0 (30 min), 1–4, and 14 p.i. (C).

correlate with the peak of shedding of *Leptospira* in urine (days 12–15) (Fig. 2D). Following lethal infection with $10^8$ *L. interrogans*, 100% of mice met endpoint criteria between days 8–11 postinfection (Fig. 2A) which was preceded by a significant loss in weight starting on day 5 (Fig. 2B). Bacteriemia peaked on day 1, starting 30 min after infection (Fig. 2C and inset), that was followed by initiation of urine shedding on day 5 (Fig. 2D). The sublethal data were consistent with our previous results (18, 19).

***Leptospira* DNA was found in mouse testes after lethal and sublethal infection.** At termination after sublethal infection (day 15), all mouse kidneys tested positive for leptospiral DNA ($\sim 10^4$ equivalent genomes/mg) which was a good measure of infectivity of the Fiocruz L1-130 strain. In testes, 3/8 (37.5%) mice tested positive for *Leptospira* after infection with $10^5$ leptospires and 6/8 (75%) mice tested positive after infection with $10^6$ leptospires, although all leptospiral loads were very low, at the limit of qPCR detection (Fig. 3A, Table 1). In contrast, after lethal infection with $10^8$ leptospires, at 30 min p.i, 3/3 mice had 30–1000 leptospires per mg of testes tissue, and after 24h postinfection 40–740 leptospires per mg of tissue were detected in 3/3 mice (Fig. 3B). In another experiment 6/13 (46.15%) mice had very low numbers of leptospires in the testes at 8–11 days p.i (Fig. 3B, Table 1). As expected, higher leptospiral load was detected in kidneys compared to testes (Fig. 3A, B, Table 1). Lethal and sublethal infection with *L. interrogans* (Table 1) led to spirochetal accumulation measured by qPCR in the kidney tubules of 100% of mice.

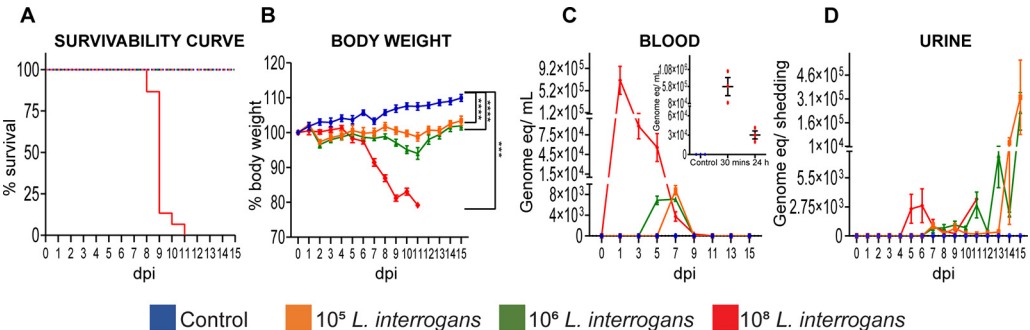

**FIG 2** Weight, survival differences and dissemination in body fluids after lethal and sublethal infection of mice with *L. interrogans*. Survival of mice after sublethal and lethal (A) infection; body weight was measured daily until 15 days postinfection (B); blood was collected on alternate days and tested for *Leptospira* load by 16S rRNA qPCR (C). Urine was collected daily and tested for *Leptospira* load by 16S rRNA qPCR (D). qPCR data is represented as genome equivalent (eq) denoting number of leptospires. Blood was collected after early lethal infection at 30 min and 24 h (Data of one experiment, $n = 3$ mice per group) and qPCR was performed to determine *Leptospira* load by 16S rRNA qPCR. Sublethal infection is represented as data of two independent experiments; $n = 8$ mice. Lethal infection is represented as data of three independent experiments; $n = 13$ mice. Statistics was performed by *unpaired t test* with Welch's correction between control and infected groups. dpi, day postinfection, ***$P < 0.001$, ****$P < 0.0001$.

**Live *Leptospira* was cultured from testes after sublethal and lethal *L. interrogans* infection.** Morphology and motility of *L. interrogans* was checked under a dark field microscope and confirmed by qPCR in cultures from kidney and testes of mice. For sublethal infections, in mice euthanized 15 days p.i, we recovered live *Leptospira* in cultures from 1/8 (12.5%) testes of mice infected with $10^5$ *Leptospira* and from 2/8 (25%) testes of mice infected with $10^6$ *Leptospira* at 3 and 5 days after inoculation of the tissue into EMJH media. The *Leptospira* were motile but presented in dot form, not in spiral form. Positive dark field cultures were confirmed to species by 16S rRNA qPCR (Fig. 4A and Table 2). For lethal infections, we recovered ~1750 live *Leptospira* in 3 day testes cultures from mice euthanized 30 min postinfection, 500 *Leptospira* were detected in 3 day cultures of mice euthanized 24h postinfection and no *Leptospira* was recovered from mice euthanized 8–11 days postinfection. Kidney cultures, used as a positive control for infection and tissue colonization, were positive in 12.5–37.5% of mice infected with $10^5$ and in 37.5–87.5% of mice infected with $10^6$ *Leptospira* (motile spirochetal forms) on days 3 and 5 after inoculation of the tissue into EMJH media,

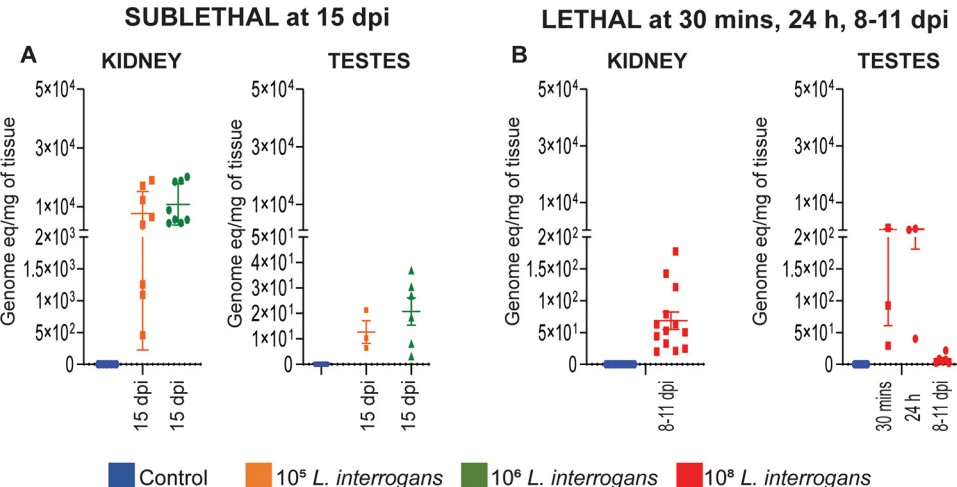

**FIG 3** *Leptospira* presence in testes and kidney after sublethal and lethal infection. Testes and kidney were harvested at termination at day 15 (sublethal) and 8–11 (lethal) and *Leptospira* load was measured in the tissues by 16S rRNA qPCR. qPCR data is represented as genome equivalent (eq) denoting number of leptospires. Sublethal (A), $n = 8$ mice, two independent experiments; lethal 8–11 dpi (B), $n = 13$ mice, data represents three independent experiments, lethal 30 min and 24 h (B), $n = 3$ mice, data represents one experiment. dpi, day postinfection.

**TABLE 1** Tissues positive for *Leptospira interrogans* 16s rRNA after lethal and sublethal infection[a]

| Kidney | | | Testes | | |
|---|---|---|---|---|---|
| LIC genome eq/mg tissue No. of positive/total (%) | | | LIC genome eq/mg tissue No. of positive/total (%) | | |
| | | 8–11 dpi | 15 dpi | | 8–11 dpi |
| 15 dpi | | | | | |
| $10^5$ | $10^6$ | $10^8$ | $10^5$ | $10^6$ | $10^8$ |
| 8/8 (100) | 8/8 (100) | 13/13 (100) | 3/8 (37.5) | 6/8 (75) | 6/13 (46) |

[a]Data represents *n* = 8 mice, for sublethal infection, data of two independent experiments and *n* = 13 mice, for lethal infection, data of three independent experiments. dpi, day postinfection.

whereas no *Leptospira* was cultured from kidneys from mice infected with the lethal dose (Fig. 4B and Table 2).

**Sublethal and lethal infection with *L. interrogans* did not cause inflammation in testes.** No mononuclear cell infiltration or tissue structure morphological changes were observed in testes tissues from mice infected with $10^5$, $10^6$ and $10^8$ *Leptospira* after H&E staining, whereas histopathology analysis of kidney showed increased infiltration of immune cells in *L. interrogans* infected mice compared to control (Fig. 5).

## DISCUSSION

Analysis of dissemination of *Leptospira* to sex organs in mice is of significant value as it queries the possibility of mammal-to-mammal venereal transmission. Previous reports regarding dissemination of *Leptospira* to sex organs in mares, cattle, boars, sheep and rams have suggested that leptospirosis may be sexually transmitted in large animals (5–11, 13, 14). This leads to reproductive failure and pregnancy issues like abortion, stillbirth, and sick offspring (6–8). In this study, we used two serovars of pathogenic *Leptospira*, two strains of mice (C57BL/6 immunocompetent and immunocompromised C3H/HeJ with a mutation in TLR4) and two different techniques to monitor *Leptospira* dissemination after infection to determine if live *Leptospira* disseminates to and colonizes the testes.

Both $10^5$ and $10^6$ sublethal doses of *L. interrogans* serovar Copenhageni led to survival of all male mice whereas the $10^8$ dose led to death of all infected animals before the term of the experiment (15 days). Both sublethal and lethal infections led to dissemination of *Leptospira* to blood, urine, kidney and testes (Fig. 2 and 3). A $10^8$ dose of *Leptospira* had been used previously in our lab to induce sublethal infection of mice (20). Important differences between these experiments are the route of infection and the sex of the mice. In the first study we used female mice infected via eye drop (20).

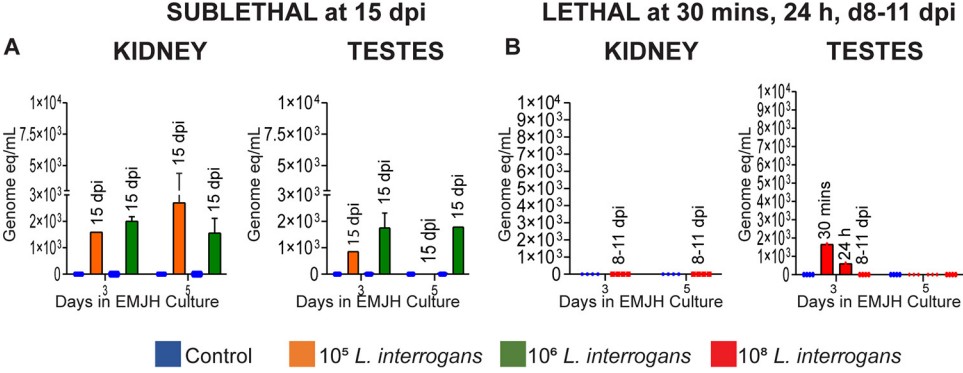

**FIG 4** Live *Leptospira* was recovered in EMJH culture of testes and kidney after sublethal but not lethal infection. 2 μL of EMJH testes and kidney culture was tested by 16S rRNA qPCR on days 3 and 5 after termination at day 15 (sublethal) and days 8–11 (lethal). qPCR data is represented as genome equivalent (eq) denoting number of leptospires. Panel A, *n* = 8 mice, data represents two independent sublethal experiments and Panel B *n* = 4 mice, data represents one of two independent lethal experiments (8–11 dpi), *n* = 3, data represents one lethal experiment (30 min and 24 h). dpi, day postinfection.

**TABLE 2** PCR confirmation of *Leptospira interrogans* in EMJH culture of testes and kidney after lethal and sublethal infection[a]

| | Kidney | | | | Testes | | | |
|---|---|---|---|---|---|---|---|---|
| | LIC genome eq/mL | | | | | | | |
| | | *L. interrogans* | | | | *L. interrogans* | | |
| | | 15 dpi | | 8–11 dpi | | 15 dpi | | 8–11 dpi |
| Days in culture | Control | $10^5$ | $10^6$ | $10^8$ | Control | $10^5$ | $10^6$ | $10^8$ |
| | No. of positive/total (%) | | | | | | | |
| Day 3 | 0/8 (0) | 1/8 (12.5) | 4/8 (50) | 0/8 (0) | 0/8 (0) | 1/8 (12.5) | 2/8 (25) | 0/8 (0) |
| Day 5 | 0/8 (0) | 2/8 (25) | 4/8 (50) | 0/8 (0) | 0/8 (0) | 0/8 (0) | 1/8 (12.5) | 0/8 (0) |

[a]Presence of *Leptospira* was confirmed by 16s rRNA qPCR after placing the kidney and testes in EMJH culture and tested at days 3 and 5, $n = 8$ mice, data of two independent experiments. dpi, day postinfection.

In the current study, as we were aiming to evaluate the presence of *Leptospira* in testes after sublethal and lethal infection, we inoculated male mice with a $10^8$ dose of *Leptospira* as we had previously observed higher susceptibility to infection in male hamsters (16). We used an intraperitoneal route for infection as we had previously observed that male mice infected with an IP $10^8$ dose had *Leptospira* in blood 24h day after infection (21). Furthermore, the IP route is the only route that allows us to administer a predetermined infectious dose to induce sublethal or lethal infection. A comparison of disease progression markers between lethal and sublethal infections in male mice led to the interesting observation that lethal infection leads to a much higher *Leptospira* load ($\sim 7 \times 10^5$) in blood 24h postinfection than sublethal infection which only peaked on days 5–7 at $\sim 10^4$. The latter was observed in our previous sublethal studies (18, 19) consistent with what has been shown in the bioluminescent model, where the peak of MFLum1 dissemination in blood at days 3 and 4 was proportional to the infectious dose, and also proportional to the level of renal colonization 1 month p.i (17). This was also observed by others in rats (22). In addition, after lethal infection body weight started to decline on day 5 as *Leptospira* colonizes the kidney and the mice kept losing weight ($>$10% on day 7) reaching the 20% endpoint between days 8–11, whereas after sublethal infection, mice lose $<$ 10% of weight until day 11 after which they start to recover as reflected by weight gains and increased *Leptospira* shedding in urine on days 12–15. Increased shedding of *Leptospira* in urine by otherwise healthy mice signals successful kidney colonization which ensures that *Leptospira* continues to be shed into the environment and completes the enzootic cycle. Detection of high burdens of *Leptospira* in blood 24h after infection followed by a 10% weight loss and shedding of a low burden of *Leptospira* in urine within the first week of infection (days 5–7) predicts worse clinical outcomes, as also shown recently for C57B/6 mice (23, 24). These differences in kinetics of disease progression between lethal and sublethal infections in mice are important to design better diagnostic assays that can detect leptospirosis earlier and predict disease outcomes.

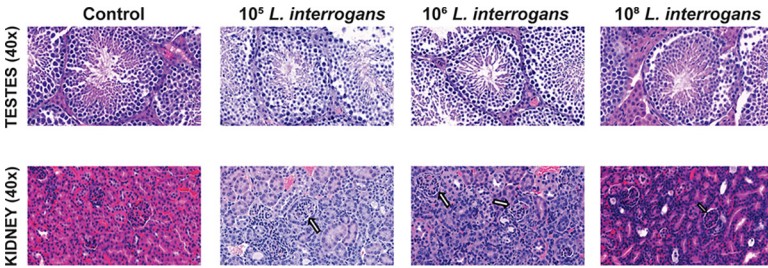

**FIG 5** *L. interrogans* infection led to renal inflammation without testicular inflammation. Representative images of H & E staining for testes and kidney. Images were captured at $\times$40 magnification. Arrow indicates increased immune cell infiltration in kidney sections, data represents one of two independent experiments for lethal infection ($10^8$ LIC); for sublethal infection ($10^5$ and $10^6$ LIC) data represents one experiment.

*Leptospira* was detected in testes of 18/18 mice as early as 30 min post lethal infection and in 19/19 mice 1–4 days postinfection by DNA qPCR and bioluminescence, respectively (Fig. 1, 3). Whether this shows a tropism of leptospires for testis or if this a passive effect due to the blood dissemination and high vascularization of testis is an interesting question that remains to be studied. Low numbers of *Leptospira* DNA were still detected at 8–15 days postinfection after lethal and sublethal infections in 37.5–75% of mice (Fig. 1, 3). The data conclusively shows presence of *Leptospira* in mouse testes and that the load of *Leptospira* in testes decreases as the infection progresses in time (Fig. 3). This is consistent with the bioluminescent C57BL/6 model, showing that leptospires disappear from circulation after 1 week p.i to progressively reappear in one organ, the kidney (17). Previous studies have detected Leptospiral DNA in vaginal fluid of cows (10), ovaries, uterus and uterine tubes of sheep (11, 12). Although we did not study the presence of leptospiral in these organs, we can speculate that leptospires may also infect the reproductive tract of female mice.

Another important finding in this study was detection of live *Leptospira* in testes of 12.5–25% of mice, 15 days after sublethal infection. Morphology and motility were checked under a dark field microscope. *Leptospira* were motile but did not present in spiral form and positive dark field cultures were confirmed to species by 16S rRNA qPCR. We considered mobile non spiral *Leptospira* viable as these forms often present in culture and then later morph into spirochetal forms if given enough time and suitable culture conditions. Viability of *Leptospira* in testes is further supported by bioluminescence of *Leptospira* visualized in testes of 19/19 mice up to 4 days postinfection with *L. interrogans* serovar Manilae. In contrast, *Leptospira* was not recovered in EMJH cultures of testes 8–11 days after lethal infection, despite renal colonization. We hypothesize that the lack of oxygen in the organs of moribund mice could have harmed or killed the remaining leptospires, that are aerobic bacteria. Earlier studies also determined viability of *Leptospira* in culture from testes of 13.63% infected boars and 41.67% infected rams (6, 14). Reports suggest that venereal spread of *Leptospira* from male to females is plausible. Cilia et al. stated that the presence of serovar Bratislava in boars could be transmitted to sows during mating season (6) and a similar probability of transmission in ovines has been reported (25, 26). *Leptospira* has been detected in semen (27–30) and oviducts (7, 31) of cattle. Reproductive problems have been encountered due to genital leptospirosis like early fetal death (8, 32), abortion (33, 34), lower fertility (34), estrus repetition (35) in mares, swine and cattle. The above-mentioned issues may occur due to presence of *Leptospira* in semen, in turn affecting the sperm viability. Alteration in spermatogenesis due to damage, affected sperm viability in 3.5% infected bulls, with no molecular detection by PCR, in the same study, viable sperm was observed with molecular detection in 1.5% infected bulls (36). These data suggest that sexual activity may be a source of transmission in infected individuals.

We also found that presence of live *Leptospira* in testis of infected mice did not result in inflammatory changes to this organ. This was also observed in boars infected with pathogenic *Leptospira* (6, 37). In another study, experimental infection with *Leptospira pomona* showed no testicular lesions in 33.33% bulls (13). No inflammatory triggers were observed after exposure of bovine endometrial cells to heat killed *Leptospira* and outer membrane Leptospiral proteins (38). We speculate that viable forms of *Leptospira* present in testes may not cause tissue damage which may allow them to remain undetected and a possible source of transmission.

In this study we show presence of live *Leptospira* in testes of mice as early as 30 min postinfection and up to 4 days postinfection in 100% of the mice tested, which is within the spirochetemic phase of infection. Given that a much lower percentage of individuals (up to 25%) were infected up to 15 days postinfection with low numbers of *Leptospira* 7 days after *Leptospira* exits the blood phase, the data suggests that colonization of testes may be transient and mostly limited to the spirochetemic phase of infection and that transient colonization is insufficient to cause histopathological changes. Further studies are necessary to evaluate if presence of *Leptospira* in testes of

some mice leads to excretion in semen and to venereal transmission using either artificial insemination of females with spiked semen or other methods that allow for bypassing possible contamination of semen with urine of the same mouse.

## MATERIALS AND METHODS

**Animals.** Two- to 6-month old female and male albino C57BL/6J Tyrc-2J mice (bred in the animal facility at Institut Pasteur) and 8-week old male and female C57BL/6J mice (Janvier Labs, France) were used in this study. All protocols were reviewed by the Institut Pasteur (Paris, France), the competent authority for compliance with the French and European regulations on Animal Welfare and with Public Health Service recommendations. This project has been reviewed and approved (#2014-049) by the Institut Pasteur ethic committee (CETEA #89). C3H-HeJ male mice, 7-weeks old, were purchased from The Jackson Laboratory (Bar Harbor, ME) and maintained in a pathogen-free environment at the Laboratory Animal Care Unit of the University of Tennessee Health Science Center (UTHSC). All experiments were performed in compliance with the UTHSC Institutional Animal Care and Use Committee Protocol no. 19-0062 (18, 19).

**Bacteria.** Pathogenic *Leptospira interrogans* serovars Manilae and Copenhageni (strain Fiocruz L1-130) were cultivated in EMJH media, resuspended in PBS and spirochetes were enumerated by dark-field microscopy (Zeiss USA, Hawthorne, NY), using a Petroff-Hausser counting chamber, as previously described (39). For infection of C3H/HeJ mice, bacteria retrieved from infected hamsters were used after 2 culture passages. Two bioluminescent strains of *L. interrogans* obtained after chromosomal insertion of the firefly luciferase cassette by random mutagenesis using the Himar1 transposon were used for live imaging. MFLum1 is a derivative of *L. interrogans* serovar Manilae strain L495 (17) and FGLum4 is a derivative of *L. interrogans* serovar Copenhageni Fiocruz L1-130. MFLum1 is a virulent strain able to provoke severe leptospirosis and renal colonization (17) whereas FGLum4 is a mutant strain in LIC10006 (33810.1), the DNA gyrase subunit A (gyrA), and is unable to colonize the mouse kidney.

**Infection of mice and study design.** For live imaging, groups of 4 to 5 albino or black C57BL/6J females and males were intraperitoneally injected with PBS (control) or with $5 \times 10^7$ or $2 \times 10^8$ *L. interrogans* MFLum1, or $2 \times 10^8$ *L. interrogans* FGLum4 in 200 $\mu$L of PBS (infected). Live imaging of mice was performed as in (17) and recently described in detail (23). Detection and kinetics of live leptospires dissemination was done by tracking the bioluminescence 10 min after injection of Luciferin (30 mg/mL in PBS).

For the C3H lethal/sublethal infection study, groups of C3H/HeJ ($n = 3$ to 4) were injected with PBS (control) and with $10^5$, $10^6$ (sublethal doses) or $10^8$ (lethal dose) *L. interrogans* serovar Copenhageni intraperitoneally. Survival and body weight was monitored daily, urine was collected daily, and blood was collected on alternate days. Baseline fluid collection was done on day 0. At termination on day 15, kidney and testes were collected from both groups of mice for quantification of *Leptospira* load, for culture determination of bacterial viability and tissues were stored in 10% formalin for H&E staining (18, 19). For lethal infection, the endpoint criteria was weight loss reaching 20% or depressed state (ruffled fur and loss of mobility) with >15% weight loss.

**Leptospira detection by q-PCR.** DNA isolation from blood, urine, kidney and testes sections were carried out using NucleoSpin tissue kit (Clontech, Mountain View, CA) according to manufacturer's instructions. *Leptospira* 16S rRNA TAMRA probe CTCACCAAGGCGACGATCGGTAGC, forward primer CCCGCGTCCGATTAG and reverse primer TCCATTGTGGCCGAACAC (Eurofins Genomics, Huntsville, AL) were used for *Leptospira* detection using qPCR (18, 19). A standard curve of $10^5$ to 1 *L. interrogans* was run with the samples for quantification. At termination, kidney and testes were placed in EMJH culture to determine viability (motility and morphology) of by dark field microscopy (20X, Zeiss USA, Howthorne) and *Leptospira* was quantified in culture supernatant by 16S rRNA qPCR on days 3 and 5 post tissue inoculation.

**Histopathology by HandE staining.** Kidney and testes tissues were fixed in 10% formalin buffer, paraffin embedded, sectioned and stained by H&E. Histopathology was performed at the Core Histology Department, UT Methodist University Hospital, Memphis, TN. Images were captured using CaseViewer software after digitally scanning the tissues section using Pannoramic 350 Flash III (3D Histech, Hungary).

**Statistical analysis.** Graphs were plotted using GraphPad Prism software. *Unpaired t test* with Welch's correction was used to analyze differences between control and *L. interrogans* infected groups, $P < 0.05$ is significant.

## ACKNOWLEDGMENTS

We thank the Histology department from UT Methodist University Hospital for processing and staining tissues, Michelle Morrison from the Department of Pathology, UTHSC for digitally scanning tissue slides and Sheila Criswell at the Department of Diagnostic and Health Sciences, UTHSC for analysis of testes sections. This work was supported by the Public Health Service awards AI139267 (M.G.S.), AI142129 (M.G.S.), and AI155211 (M.G.S.), from the National Institute of Allergy and Infectious Diseases (NIAID) of the National Institutes of Health (NIH) of the United States of America, and by a Ph.D fellowship "DimMalinf" to C.W. for Gwendoline Ratet. The content of this manuscript is

solely the responsibility of the authors and does not necessarily represent the official views of NIAID or NIH.

Conceptualization: Advait Shetty, Maria Gomes-Solecki, Frédérique Vernel-Pauillac, Catherine Werts. Experimental investigation: Advait Shetty, Suman Kundu, Frédérique Vernel-Pauillac, Gwendoline Ratet. Data analysis: Advait Shetty, Maria Gomes-Solecki, Frédérique Vernel-Pauillac, Catherine Werts. Writing original draft: Advait Shetty, Maria Gomes-Solecki, Catherine Werts. Editing: Maria Gomes-Solecki, Frédérique Vernel-Pauillac and Catherine Werts. Supervision and funding acquisition: Maria Gomes-Solecki, Catherine Werts.

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
