## [Reviewer comments · Microbiology Spectrum]

Microbiology Spectrum

Transient Presence of Live *Leptospira interrogans* in Murine Testes

Advait Shetty, Suman Kundu, Frederique Vernel-Pauillac, Gwendoline Ratet, Catherine Werts, and Maria Gomes-Solecki

Corresponding Author(s): Maria Gomes-Solecki, University of Tennessee Health Science Center

Review Timeline:

Submission Date:

March 16, 2022

Accepted:

March 25, 2022

Editor: Catherine Brissette

Reviewer(s): The reviewers have opted to remain anonymous.

Transaction Report:

DOI: <https://doi.org/10.1128/spectrum.02775-21>

March 25, 2022

Dr. Maria Gomes-Solecki
University of Tennessee Health Science Center
Microbiology, Immunology and Biochemistry
858 Madison Ave
Memphis, TN 38163

Re: Spectrum02775-21 (Transient Presence of Live *Leptospira interrogans* in Murine Testes)

Dear Dr. Maria Gomes-Solecki:

The main concern with the original paper was the hyperbolic title and language which implied sexual transmission; this was removed. Additional experiments in other mouse models as well as other leptospira serovars strengthens the data presented.

Your manuscript has been accepted, and I am forwarding it to the ASM Journals Department for publication. You will be notified when your proofs are ready to be viewed.

Sincerely,

Catherine Brissette
Editor, Microbiology Spectrum
